# High Hopes for the Biofabrication of Articular Cartilage—What Lies beyond the Horizon of Tissue Engineering and 3D Bioprinting?

**DOI:** 10.3390/biomedicines12030665

**Published:** 2024-03-15

**Authors:** Yordan Sbirkov, Murad Redzheb, Nico Forraz, Colin McGuckin, Victoria Sarafian

**Affiliations:** 1Department of Medical Biology, Medical University of Plovdiv, 4000 Plovdiv, Bulgaria; 2Research Institute, Medical University of Plovdiv, 4000 Plovdiv, Bulgaria; 3MatriChem, 1113 Sofia, Bulgaria; m.redzheb@matrichem.com; 4CTIPharma Department, Cell Therapy Research Institute, CTIBIOTECH, 69330 Lyon, France; nico.forraz@ctibiotech.com (N.F.); c.mcguckin@ctibiotech.com (C.M.)

**Keywords:** articular cartilage, 3D bioprinting, tissue engineering, regenerative medicine

## Abstract

Technologies and biomaterials for 3D bioprinting have been developing extremely quickly in the past decade as they hold great potential in tissue engineering. This, together with the possibility to differentiate stem cells of different origin into any cell type, raises the hopes in regenerative medicine once again after the initial breakthrough with stem cells in the 1980s. Nevertheless, three decades of 3D bioprinting experiments have shown that the production of functional tissues would take a longer time than anticipated. Cartilage, one of the simplest tissues in the body, consists of only one cell type. It is not vascularised and innervated and does not have lymphatic vessels either, which makes it a perfect target tissue for successful implantation. The tremendous amount of work since the beginning of this century, combining the efforts of bioengineers, material scientists, biologists, and physicians, has culminated in multiple proof-of-concept constructs that have been implanted in animals. However, there is no single reproducible, standardised, widely accessible and accepted strategy that can be readily applied in the clinic. In this review, we focus on the current progress in the field of the 3D biofabrication of articular cartilage and critically assess failures and future challenges.

## 1. Introduction—Complexity of Hyaline Cartilage and Unsatisfied Clinical Needs

Hyaline cartilage forms during embryogenesis, and in adults, it is found in the joints (articular cartilage), the nasal septum, the trachea, vertebral discs, and other locations. It presents as a unique tissue in many aspects. First, it has no blood supply or nerve endings. Chondrocytes are entirely dependent for nutrition on synovial fluid on one side and on diffusion from the bone (and its blood supply) they cover on the other side. There is only one type of very sparsely interspersed cell (only ~2% of the total mass) that builds hyaline cartilage—chondrocytes. However, they have distinct properties and build different types of extracellular matrix (ECM) networks depending on the zone they are located in. In the superficial zone (10–20% of the total thickness), reside cells, which are flatter, more densely packed compared to the other zones, and have more progenitor cell-like properties. They are less differentiated and preserve their potential to divide and have some regenerative capacity. These are the so-called articular cartilage-derived progenitor cells (ACPCs). A subpopulation of cells similar in function are the cells found in the perichondrium (fibroblast-like cells/perichondrocytes with chondrogenic potential and chondroblasts) in the nasal cartilage [1,2]. Interestingly, these cells have been extensively used to regenerate knee injuries as autografts [3].

In the transitional zone (40–60% of the cartilage), cells are round-shaped and arranged in a columnar fashion perpendicularly to the superficial zone [4]. In the deep zone (~30% of the cartilage), the cells are progressively more hypertrophic and have exited the cell cycle. In the calcified zone, which provides the integration of the cartilage with the subchondral bone surface, there are sparsely dispersed hypertrophic chondrocytes. Perhaps the most significant difference between the latter three zones is the composition of the ECM. In the transition zone, collagen fibres are randomly oriented, while in the deep zone, they appear to be parallel to each other and perpendicular to the superficial zone. In the calcified zone, there is less collagen (~20%) and more ossification [5]. This unique arrangement of cells and ECM components (mainly different types of collagens, proteoglycans, and 65–80% water) provides the required mechanical load-bearing properties of this type of connective tissue [6,7].

Chondrocytes have little to no regenerative capacity, which means that once damaged, hyaline cartilage cannot be naturally replenished. If just osteoarthritis (OA) is considered, there are over 240 million people worldwide with symptomatic disease [8]. Interestingly, there have been various strategies related to tissue engineering or using acellular biomaterials to repair damaged cartilage or at least to alleviate the symptoms, but 3D bioprinting has not found its way to the clinic yet.

## 2. Sources of Patient-Derived Cells for Cartilage Tissue Engineering—Chondrocytes, iPSCs, Bone Marrow—Or Adipose-Derived Stem Cells?

### 2.1. Cartilage

Historically, the attempt to use autologous cartilage for transplantation has been in trials the longest. Nevertheless, there is no standardised and well-accepted solution for the problem of expanding chondrocytes in vitro. This presents a critical issue in tissue engineering that we have failed to tackle so far in the past three decades. Nevertheless, there has been significant progress in that field. A great amount of work has focused on how to make primary chondrocytes expand in vitro without losing their properties. Jiang et al. demonstrated that cartilage stem/progenitor cells (CSPCs) are in fact a dynamic subpopulation that can emerge under cell culture conditions and can be expanded. The authors used femoral condyle chondrocytes and grew them at low density in monolayers and with low glucose to observe the increase in the stem cell marker CD146, decrease in collagen II, and enhanced proliferation in about 25% of the total cell population by day 5 of culture/post isolation [9]. Jiang et al. also showed that while CD146 positive cells maintain a high proliferative and differentiation capacity in low density, these stem cell-like properties are decreased in conditions of low glucose 2D cultures, high cell density, and growth in 3D micromasses. Therefore, an optimal strategy for tissue engineering and the 3D bioprinting of cartilage would be to expand these CD146 high, Col II low cells in standard monolayer conditions until sufficient numbers are generated for printing (7–10 million cells per cm^3^ based on Hunziker et al. [10]). However, this approach has been reported by only one study so far [11].

Chondrocytes, together with their surrounding pericellular matrix (PCM), form the so-called chondrones, which have been suggested to possess superior mechanical and biosynthetic properties than isolated single chondrocytes [12]. In an animal model, growing primary chondrocytes together with isolated chondrones significantly enhances the secretion of aggrecans and collagen II in vitro and improves repair processes in the knee when cells are encapsulated in alginate spheres (6 weeks) and implanted. The authors detected improved Wakitani scores (a histological scoring of the repaired tissue) and concluded that both cell types must be included in regenerative therapies [12]. 

Perhaps the most successful and clinically relevant work so far is based on the isolation and expansion of nasal chondrocytes (NCs) in vitro. After 2 weeks, the cells were seeded onto collagen I/III membranes at ~20 million cells/cm^3^ and grown in a GMP facility for another 2 weeks before implantation. Even if patients were relatively young and active (median age of 38 years) and there was some heterogeneity in the amount of ECM deposition between patients (as judged by the GAG and Col II staining of the membranes), the autologous graft failed in only one patient. In four patients (out of nine patients, treated for a total of 11 defects, followed up for 24 months), there was 100% success in filling in the original knee defects [13]. Importantly, the authors also assessed by MRI the glycosaminoglycan, water, and collagen II content of the repaired tissue compared to normal cartilage and confirmed a very similar composition to normal healthy joints. Later, the authors demonstrated that the proliferation and chondrogenic capacity in the culture of debrided knee chondrocytes is inferior to that of NCs, obtained from the same patients, resulting in lower amounts of the cartilage-specific matrix components glycosaminoglycans and type II collagen [14]. The successful development of NC-based articular cartilage regeneration therapy has been continued by the same group in the context of two clinical trials (NCT01605201 and NCT02673905). A recent report demonstrated a GMP-compliant substitution of autologous serum, used in earlier studies, with a clinical-grade human platelet lysate [15]. Nevertheless, the spatial organisation of the cells and ECM and the functionality of the cartilage engineered in this way have not been assessed in detail. 

### 2.2. Stem Cells—iPSCs, BMSCs, and ADSCs

Wu et al. proved using primary chondrocytes, a histological examination of developing limbs (even growth plates of 60-year-old adults), and PSCs that pre-chondrocytes/progenitor cells can be distinguished by the high expression of BMPR1B, LIFR, and CD146 and the low presence of CD166. In accordance with what Jiang and colleagues found, mature or hypertrophic chondrocytes which are not dividing anymore show low CD146 expression. Therefore, monitoring these markers during cartilage engineering from iPSCs or PSCs may be a useful approach [16]. 

Stem cells have long been regarded as the most promising tool in regenerative medicine in general, not just in cartilage tissue engineering. Various protocols have been optimised and validated numerous times with bone marrow-derived mesenchymal cells. A good example of how chondrogenesis protocols can combine differentiation media, scaffolds harvesting certain components of the natural ECM of cartilage, and biomechanical stimuli is provided by Lin et al. [17]. The authors developed a protocol with the sequential exposure of MSCs to a chondrogenic medium (TGF-β, dexamethasone, ITS, ascorbate, pyruvate, and proline), normal growth medium (1 week each), and then culture on methacrylate 3D scaffolds enriched with hyaluronic acid (HAMA) that undergo/experience dynamic compression for 14 days (again in chondrogenic medium). The resulting 3D constructs demonstrated excellent staining for collagen II and with Safranin O and toluidine blue, as well as the enhanced expression of *Aggrecan* and *Sox-9* all of which suggest successful chondrogenesis by the abovementioned conditions [17]. These conditioned MSC-laden HAMA scaffolds were then implanted in mice to repair pre-induced knee-joint defects for 8 weeks and showed a good recovery of the lesion areas. Nevertheless, 2 months post-implantation, there were almost no MSC-derived cells remaining within the repaired section [17]. 

The importance of the temporal and spatial (microphysiological) factors like the timing of chondrogenic induction, the diffusion of the growth medium, and hypoxia throughout the engineered tissue has been highlighted by Futrega et al. The authors demonstrated using TGF-β1-stimulated BMSCs and RNA sequencing at different time points (day 1, 3, 7, and 21) after the initiation of chondrogenic differentiation by the growth factor, that in fact a single day of exposure to TGF-β1 is sufficient to activate chondrogenesis gene expression programmes at the examined timepoints. According to this study, the results can be achieved only in micromasses of around 5000 cells rather than in macromasses of tens of thousands of cells, which have been the gold standard since 1998 [18], and which leads to heterogeneous chondrogenic induction [19]. Therefore, the starting number of cells that are being induced to differentiate should also be considered, keeping in mind that several million chondrocytes (which have poor proliferative capacity if differentiated adequately) would be needed in the end.

Interestingly, adipose tissue is richer in MSCs than bone marrow. These MSCs have high telomerase activity, meaning that they could be expanded in vitro more efficiently, and can also be differentiated into chondrocytes. Furthermore, lipoaspiration as a technique is less painful than the acquisition of bone marrow aspirates for BMSCs. Taken together, these make adipose-derived MSCs an attractive starting material for cartilage tissue engineering, and these cells and their products are currently in clinical trials for OA treatment with promising results [20,21,22]. 

## 3. The State of the Art in Bioprinting Technologies and Strategies for Cartilage Biofabrication

Three-dimensional bioprinting has several advantages that are unsurpassed by any other current technology that can be implicated in tissue engineering—speed, precision, control, and the reproducibility of the spatial organization and the composition of both cells and ECM components. While the biomaterials and resolution of 3D bioprinting have been constantly improving for the past 20 years since the first bioengineered cartilage in vitro models [23,24], there are no personalised cartilage implants with satisfactory functional biology and mechanical properties yet. Nevertheless, there are a few examples where 3D bioprinting has been applied in an attempt to answer, at least partially, certain cartilage tissue engineering questions.

### 3.1. Have We Found the Best Natural and/or Synthetic Polymers for Cartilage Tissue Engineering?

The very first studies on the vast possibilities bioprinting can offer for cartilage regenerative medicine started nearly 20 years ago with simpler biomaterials based on alginate [25] or polyethylene glycol (PEG) [23]. Since then, a plethora of hydrogels and biomaterials have been tested in various combinations. Regarding synthetic polymers, for instance, Cui et al. used inkjet bioprinting, a bioink consisting of human chondrocytes and polyethylene glycol diacrylate (PEGDA), and photopolymerization that resulted in over 60% cell viability, which from today’s perspective, is not impressive. Nevertheless, proteoglycan deposition gradually increased within the constructs (assessed by the Safranin O staining of bioprints up to 4 weeks post printing) demonstrating good compatibility of this hydrogel with human chondrocytes [26]. 

With regard to natural polymers, alginate has been used extensively in the first decade of cartilage bioprinting [25,27,28] and has been mixed with other polymers like gellan and ECM particles from decellularised human cartilage (BioCartilage^®^). This approach showed an increased proliferation of bovine chondrocytes and a slight increase in proteoglycan and collagen production. However, the ECM components secreted by the cells were mostly stimulated by the addition of TGF-β in the medium and not by the natural ECM components found in BioCartilage^®^ [29]. Besides alginate, about a decade ago, one of the first extrusion-based (EB) 3D bioprinting studies with hyaluronic acid was carried out. The focus of this work was the combination of two tunable polymers into a biocompatible hydrogel-poly(N-isopropylacrylamide) hyaluronan (HA-pNIPAAM), which provided fast gelation at 37 °C, and methacrylated hyaluronic acid (HAMA), which was crosslinked with UV [30]. More recently, Antich et al. optimised a hydrogel made of alginate and hyaluronic acid, which was further reinforced by polylactic acid (PLA). The authors found that primary chondrocytes from OA patients were stimulated by HA to upregulate the gene expression of the transcription factor *SOX9*, collagen II, and aggrecans and maintained an excellent viability and proliferation of the cells over a 4-week period post-printing [31]. Furthermore, alginate-based bioinks have been shown to benefit from the addition of nanocellulose [32,33] or nanocellulose crystals [34], which greatly improve the printability and biomechanical properties of the hydrogels without affecting the viability of chondrocytes. Interestingly, such formulations have been shown to have the capacity to induce chondrogenesis in iPSCs 5 weeks after bioprinting as opposed to alginate/HA hydrogels demonstrating the critical role of biomaterial ink composition [33]. 

On the other hand, collagen is a key structural protein of connective tissue, providing strength and elasticity. Unlike alginate and nanocellulose, in addition to its biocompatibility, collagen plays an active role in cell adhesion, movement, and differentiation. A high concentration of 4% atelocollagen type I biomaterial ink was used to successfully extrusion bioprint costal cartilage-derived chondrocytes [35]. The authors did not report post-bioprinting cell viability, but a subsequent 40-day in vivo study in rats demonstrated the formation of cartilage-like tissue with a typical structure with groups of isogenic cells enriched in GAGs and type II collagen. In a follow up study, the biomaterial ink was supplemented with 2.5% decellularised ECM granules, derived from costal cartilage, to stimulate the chondrogenesis of ADSCs [36]. The following implantation of both MSC-containing and cell-free constructs under the skin in the withers area of rats resulted in chondrogenesis. In some cases, it was accompanied by bone formation, demonstrating the limited in vivo control over stem cells. 

Methacrylated gelatin (GelMA) is the most utilised semisynthetic polymer in bioprinting due to a unique combination of physicochemical and bioactivity characteristics. GelMA solutions can independently be gelled physically (thermally, by cooling) and chemically (light-induced radical chain polymerization) while preserving bioactive adhesion sites and biodegradability. Given that GelMA bioinks require light-induced crosslinking (typically UV or high energy visible blue light are employed) to be stabilised at physiological temperatures, studies establishing the cyto- and genotoxicity of various aspects of GelMA bioprinting are indispensable [37]. Ruiz-Cantu et al. have established for the biomaterial inks they have formulated (15 and 20% GelMA, 0.5% *w*/*v* Irgacure 2959) that 0.62 J/cm^2^ is the minimal UV dose required to crosslink them. They observed no cyto- or genotoxicity in articular chondrocytes for a UV dose of up to at least 1.86 J/cm^2^. Unfortunately, the genotoxicity of the photopolymerization process was not reported, which is notable since the radicals produced by the photoinitiator have been proven to be cytotoxic [38,39]. Still, the authors demonstrated that bioprinting a 20% GelMA bioink containing articular chondrocytes and incubating them in vitro for 50 days resulted in a characteristic Safranin O and collagen II staining. Apart from that, the Young’s modulus increased from 200 kPa post bioprinting to 800 kPa, which is comparable to that of lateral nasal cartilage and that of the knee joint cartilage [40]. 

The popularity of GelMA biomaterial inks is also related to their capacity to achieve a wide range of stiffness values and degradation rates by varying a number of parameters including the methacrylation degree, polymer and photoinitiator concentrations, and irradiation dose. The stiffness control is especially relevant for articular cartilage engineering due to its well-known stiffness gradient from the surface to the deep zone [41]. Apart from the stiffness, the quality of the hyaline cartilage also depends on the hydrogel’s degradation rate. For instance, more compliant and faster degrading GelMA (modulus of about 8 kPa and ≈40% remaining after 28 days in culture) results in higher quality hyaline cartilage in vitro compared with stiffer and slower degrading GelMA (modulus of about 35 kPa and ≈60% remaining after 28 days) [42]. However, it remains unclear if and how these in vitro observations translate to in vivo events during cartilage repair. 

GelMA is also the most popular biomaterial for use in in situ bioprinting. This novel approach allows surgeons to simultaneously deliver bioscaffolds and cultured cells into cartilage defects in a single surgical procedure. Di Bella et al. developed a handheld device capable of co-axial extrusion and the simultaneous photocuring of the ink that is being extruded [43]. Using a GelMA/hyaluronic acid-based biomaterial ink, the authors formulated a bioink containing adipose-derived MSCs but without a photoinitiator and a cell-free ink with VA-0860 photoinitiator. The core-shell co-axial extrusion of the bioink and the photosensitive ink, respectively, ensured that the cells were protected during the curing process. When compared to microfracture and bench-based printing, the in situ bioprinting resulted in the better regeneration of the articular cartilage in sheep after 8 weeks, compared to that in the untreated control group. However, the bench-based 3D printing was not performed by co-axial extrusion and the cells were seeded on the 3D-printed scaffold rather than bioprinted. This is a significant difference given the fact that cells seeded on a hydrogel are not as uniformly distributed spatially as they are when bioprinted. Another drawback of the reported methodology is the use of UV light, which while less harmful on the hydrogel-embedded cells, can cause significant cell death in the healthy tissue surrounding the defect in vivo [44]. Thus, for the intraoperative administration of bioinks, the visible light-mediated crosslinking at 405 nm is more clinically relevant. Furthermore, in situ bioprinting will benefit enormously from its robotic-assisted implementation. 

Finally, using an articular cartilage-derived decellularised ECM (dECM), relying on the tissue native biopolymers and growth factors, is another strategy to formulate natural chondrogenic biomaterial ink. A bioink consisting of 5% GelMA, 2% alginate, 5% cartilage dECM (porcine origin), and infrapatellar fat pad-derived ADSCs from rabbits was formulated to bioprint an implant for articular defect repair in rabbits. The authors reported better cartilage regeneration for the bioprinted implant when compared to an injection of the cellular suspension alone or the implantation of cell-free 3D-printed implant followed by the injection of ADSCs [45]. 

### 3.2. Is It Possible to Engineer the Zonality, the Specific Orientation of Fibres, and the Mechanical Strength of Native Cartilage?

A number of studies have attempted to address these two important properties of hyaline cartilage—ECM fibre orientation and load bearing strength. In most, a synthetic polymer like PCL (polycarpolactone) or PEG (polyethylene glycol) was incorporated in the constructs to achieve better biomechanical characteristics but interestingly, also to guide the cells and secretion of the ECM (Figure 1).

Since alginate provided good cell viability for chondrocytes (as described previously) but little biomechanical strength, it was later reinforced with synthetic polymers like PCL. This resulted in a much better and physiologically closer compressive modulus of ~6MPa [27]. Similarly, Antich et al. demonstrated that the addition of PLA to a 2% alginate–1% HA hydrogel would yield a compression modulus of around 4MPa after 4 weeks of the culturing of primary human chondrocytes (a ~2-fold increase over the values for PLA alone) [31]. 

Another attempt to implement 3D bioprinting in order to mimic the spatial organization of hyaline cartilage was conducted in 2019. Daly et al. used the inkjet bioprinting of porcine chondrocytes and BMSCs in pre-printed PCL hydrophobic microchambers so that spheroids could form in a guided and organised manner. Perhaps the most significant achievement was that this experimental set-up could provide the spatial organization of cells needed for the orientation of collagen fibres that resembles the ones in cartilage—parallel on the surface (the superficial zone) and more randomly distributed or even perpendicular in the deeper zone. Lastly, the extrusion-based bioprinting of GelMA-embedded cells within the microchambers was also implemented by the authors showing that different tissue engineering approaches can be successfully combined [46]. 

Again using GelMA, but this time in combination with HAMA, which is known to support the growth and functionality of chondrocytes in vitro, Lam et al. demonstrated not only that cartilage can be manufactured with these materials via DLP-based resin 3D bioprinting, but also that cell density plays an important role in the production of the ECM and the self-organisation of the cells. Using 25 million cells/mL in their GelMA constructs, the authors observed that the staining for proteoglycans was more intense towards the surface, while cell numbers decreased in the deeper zones of the bioprints, which mimics the hyaline histomorphology [47].

In another attempt to recreate the zonal structure of the hyaline cartilage, Mouser et al. bioprinted heterocellular constructs using a bioink consisting of GelMA/gellan/HAMA with articular cartilage progenitor cells (ACPCs) for the superficial zone and MSCs for the middle and deep zones [48]. In spite of the promising results in terms of the chondrogenic differentiation of ACPCs and MSCs in casted biomaterial inks, bioprinting them resulted in less prominent chondrogenic differentiation accompanied by increased staining for collagen type I, a marker for hypertrophic differentiation. Thus, the shear stress generated during extrusion bioprinting needs to be optimised to achieve effective chondrogenic differentiation.

In a resin-based bioprinting approach, Shopperly et al. employed DLP to bioprint a zonally stratified cartilage construct [49]. The authors formulated three bioink formulations containing porcine articular chondrocytes and varying ratios between GelMA and HAMA, namely 1.0%:1.6%, 2.5%:1.0%, and 4.0%:0.4%, respectively. The compressive stiffness of the printed biomaterial inks increased with the relative content of HAMA from 3.1 to 5.3 kPa, which, however, is orders of magnitude lower than that of articular cartilage. Nevertheless, the in vitro investigation lasted for only two weeks resulting in data which could not provide conclusions on how successful the employed methodology was. 

Perhaps the most successful attempts so far to mimic the complex three-dimensional structure of aligned fibres are achieved by melt electrowriting (MEW), albeit using synthetic polymers such as PCL. A similar strategy to the one described above was implemented using MEW as well in 2022. This time, a mesh of orthogonal fibres with a much higher resolution (7 µm diameter) was produced using PCL and a box structure. The resulting microchambers were later loaded with porcine BMSCs via inkjet bioprinting. Upon spheroid formation within the chambers and the induction of chondrogenic differentiation, Dufour et al. characterised the morphological and biomechanical properties of these constructs. By using polarised light microscopy, the authors demonstrated the successful zonal distribution of both cells (flatter cells towards the surface) and deposited collagen fibres—tangential fibres at the surface, radial in the middle zone, and isotropic in the deeper zone. In terms of the biomechanical properties of these constructs, perhaps the most important finding was that the cells and their ECMs could enhance by more than 50 times the compressive modulus of the cell-free PCL scaffold reaching physiologically relevant values of ~400 kPa [50]. 

Castilho et al. demonstrated that having a reinforcement by a two-layer PCL scaffold, consisting of a “superficial zone” that is tangential to a “deep zone” fibre network with a box structure, provides excellent biomechanical properties to a GelMA cell-laden hydrogel. In particular, the authors recorded up to a 20-fold increase in the peak modulus when the MEW-deposited PCL network was supporting the hydrogel. Nevertheless, even the best results of ~500 kPa were still ~3-fold lower than the mechanical strength they measured of porcine cartilage. Importantly, the authors used equine chondrocytes at 20 million cells/mL but did not observe the cell density, proliferation, or deposition of large amounts of GAGs and collagen type II throughout the hydrogel to an extent that was similar to native cartilage [51]. This just demonstrates that even if new methods allow for the better resolution and control of the polymer fibre networks in engineered cartilage that provide better mechanical properties, the biological component is not at a satisfactory level yet. 

A similar strategy using PCL to reinforce the mechanical properties of the 3D-bioprinted construct was utilised by Mouser et al. They optimised a complex hydrogel made of 0.5% HAMA-pHPMA-lac-PEG (+/− PCL in each layer) for the growth and ECM (GAGs and collagen II) production of primary equine chondrocytes. Thus, while the Young’s modulus of the hydrogel alone was up to 30 kPa, when co-printed with PCL, the stiffness measured was around 8 MPa (several times higher than that of native cartilage) [52]. 

Since, however, the mechanical properties should come as a consequence of the ECM that the bioprinted cells produce rather than from synthetic polymers, few papers have measured the compressive modulus in collagen/gelatin-based constructs only. Levato et al. found that BMSCs printed in 10% GelMA significantly increase the production of the ECM with the time of the culture and reached a compressive modulus of ~150 kPa 56 days post bioprinting [11]. 

### 3.3. What Would the Best Type of Cells Be for Cartilage Reconstruction?

As discussed in the introduction, there are several different strategies to find the most appropriate cell types to recreate in vitro the complex three-layer organisation of native cartilage with its specific ECM production and fibril organisation. An excellent attempt to address this was carried out by Levato and colleagues who used equine articular cartilage progenitor cells (ACPCs), chondrocytes, and bone marrow-derived MSCs. The research group investigated different combinations of these three cell types to mimic the superficial and middle/deep zones of cartilage by bioprinting the cells in two distinct monotypic layers of GelMA. The conclusions from this study were that (1) ACPCs outperform chondrocytes in neo-cartilage formation and (2) the best model of superficial and middle zone organisation was achieved using ACPCs and MSCs, respectively, which produce distinct ECM zonal markers [11]. 

Yang et al. demonstrated using rabbit BMSCs (30 million cells/mL) that the osteochondral interface may be mimicked via extrusion-based bioprinting (at 270kPa) of bioinks made of 8% alginate, 5% gelatin (for the cartilage side) with 4% hydroxyapatite (for the bone side) and cultured for 1 week. Six months after the implantation of the constructs in pre-formed lesions in the rabbits’ knee joints, the authors observed excellent morphological features showing mainly articular cartilage filling in the lesions. Furthermore, the Wakitani scores at 6 months were 5.50 ± 2.07 compared to 9.33 ± 1.37 in the control group (scaffold without cells). Nevertheless, when load and compressive strength tests were performed, in spite of the excellent morphological data, both tests showed that the restored area possessed merely about 40% of the normal expected strength of healthy cartilage (and this was carried out in young rabbits of 3.5 months) [53].

Interestingly, Huang et al. showed that human umbilical cord MSCs (hUC-MSCs) can differentiate into chondrocytes when 3D bioprinted (extrusion) with a 10% gelatin and 5% hydroxyapatite (HAP) hydrogel. This was confirmed via a gene expression analysis for typical cartilage markers like Sox9, collagen II, and aggrecan. Importantly, the 3D bioprinted tissues were also transplanted in pig knees to fill in pre-induced defects (10 × 10 × 3 mm). The authors observed complete repair (by microscopy and standard staining protocols like H&E, toluidine blue, and Safranin O) after 6 months with the gelatin/HAP biomaterial ink. Nevertheless, no functional assessment of the restoration of the knee was carried out, nor was the ECM structure investigated in detail [54].

Table 1 summarises the current 3D-bioprinted cartilage models available. 

### 3.4. Are We Growing Biofabricated Implants in an Optimal Way?

Cell culturing conditions for MSCs or chondrocytes appear to be largely optimised, standardised between different research groups with a well-accepted gold standard of differentiation media including TGFβ, ITS, dexamethasone, ascorbic acid, etc., even with several commercially available kits. Nevertheless, growing 3D-bioprinted cartilage tissues is more complex than just having the optimal medium with growth factors for a 2D or 3D culture. The importance of bioreactors for chondrocyte differentiation and ECM production has been demonstrated in 3D spheroid cultures (spinner flask type of bioreactor) [60] and in chondrocytes seeded on scaffolds stimulated by dynamic strain [61]. In spite of the obvious advantages of bioreactors, a very small percentage of studies actually subject their 3D-bioprinted cartilage constructs to any perfusion or mechanical forces. Daly et al., using their PLC microchamber system with inkjet-printed porcine BMSCs and chondrocytes, compared the static and dynamic culturing (in a bioreactor allowing continuous perfusion) of the constructs. A positive GAG and collagen II staining after 10 weeks of culture was observed, and the perfusion system enhanced the production of these ECM components [46]. Castilho et al. subjected their PCL-reinforced GelMA constructs to dynamic load and found that GAG production by equine chondrocytes was increased ~3-fold (similarly to when cultured with TGF-β) [51]. Therefore, it appears that biomechanical stimulation may improve the quality of tissue-engineered cartilage, but a very limited amount of work (only 2 papers from >30 reviewed here) with 3D bioprinting has exploited this so far [62]. 

### 3.5. Functionalisation of Bioinks

Another important advantage of 3D bioprinting that is not, however, frequently exploited, is the opportunity to functionalise the biomaterial inks with growth factors through physical entrapment, electrostatically, or via chemical conjugation (Figure 2). 

Hydrogels can be modified by using ECM components that are naturally found in cartilage (top left), with the entrapment (top right) or affinity binding (bottom left) of growth factors (GFs), with the addition of extracellular vesicles from stem cells or microspheres containing growth factors for controlled release (bottom right). The figure was constructed using biorender.com.

An early example of the functionalisation of a biomaterial ink with TGF-β was provided by Kundu et al. The authors used a hybrid scaffold of PCL and alginate (4–6%) which successfully maintained the growth of primary nasal septum human chondrocytes. They observed that the physical entrapment of TGF-β into the bioink led to better cell proliferation and increased GAG production. The biocompatibility of these cartilage implants was tested in nude mice for 4 weeks, and a more uniform GAG staining with Alcian blue was observed showing the in vivo advantages of the functionalisation of the bioink [28]. However, no data on the release kinetics of the growth factor from the various biomaterial ink formulations were reported. 

Sun et al. utilised a mixture of gelatin, fibrinogen, hyaluronic acid, and glycerol containing GDF5 (growth/differentiation factor 5)-conjugated PLGA (50:50 PLA/PGA) microspheres to bioprint BMSCs (rabbit). This complex chondrogenic bioink was alternated with PCL to create a supporting network for these 3D constructs that were also used as implants in order to repair knee defects in rabbits. The authors demonstrated the pro-chondrogenic role of GDF5 via gene expression and histological analyses and also confirmed the repair abilities of these implants at 8 weeks [55]. In a more recent report, using a similar strategy, the authors engineered a more complex and biomimetic cartilage implant. They created a gradient in stiffness by increasing the spacing of the PLA fibres from the superficial to the deepest zone. Concurrently, a gradient in the biochemical stimulation was achieved by the incorporation of TGF-β3-conjugated PLGA microspheres in the top three-quarters of the construct and BMP4-conjugated microspheres in the deepest zone. Following a six-month in vivo study in rabbits, the implants reproduced various aspects of the zonality of the native articular cartilage, including the gradient expression of type II collagen, the superficial localisation of PRG4, and the abundant presence of type X collagen in the deep zone [56].

Imrak and Gumusderelioglu applied an interesting strategy to bioprint chondrogenic cells (the murine teratoma cell line ATDC5) in a hydrogel made of GelMA and platelet-rich-plasma (PRP). The resulting bioink was UV crosslinked initially, but the real innovation was the periodic photoactivation of PRP with polychromatic light in the near-infrared region (600–1200 nm) which enhanced the mechanical properties of the 3D-bioprinted constructs and enabled the controlled and constant growth factor release for 35 days [57]. 

Also using GelMA as the core component for the biomaterial ink, Zhu et al. found that the addition of PEGDA considerably improved the printing resolution of the constructs while increasing their compressive modulus. This work used stereolithography (SLA) for the first time to print human BMSCs, which would then differentiate (at least to a degree) into chondrocytes due to the functionalisation of the bioink with TGF-β1-loaded nanospheres. Of course, the bioprints were cultured in a standard differentiation medium containing dexamethasone, ITS, sodium pyruvate, and ascorbic acid, too [59]. 

A more biomimetic approach relies on the affinity binding of growth factors and their controlled release for optimal bioavailability and biological function. A typical strategy is to functionalise a polymer with negatively charged sulphate groups to introduce an affinity to heparin-binding growth factors. In this manner, adding a sulphated alginate to an alginate/GelMA interpenetrating network (IPN) biomaterial ink led to a greater growth factor retention over at least 21 days of the culture of the chondrogenic TGF-β3 as compared to the IPN ink without the sulphated alginate [63]. The subsequent bioprinting of bone marrow-derived MSCs and culturing over 42 days resulted in a more pronounced chondrogenesis in vitro, as judged by a more intense staining with Alcian blue and COL II expression, compared to the control. The promising results were further validated in vivo (subcutaneous implantation in mice) where the sulphated bioink with affinity-bound TGF-β3 caused significantly more cartilage ECM deposition after 4 weeks compared to a sulphated IPN bioink without exogenous TGF-β3. 

## 4. Current Limitations in 3D Bioprinting of Cartilage and Future Perspectives

The vast possibilities that the toolset of 3D bioprinting provides have already been widely recognised in the context of disease modelling. Especially with regard to cancer, drug testing, and personalised medicine, our work and that of others have demonstrated that 3D bioprinting can reproduce in vitro the in vivo biology of tumours perhaps better than any other technique. There are many examples from colorectal [64,65,66], lung [67], breast [68,69], and other types of cancer [70,71]. Furthermore, the advantages of 3D bioprinting for regenerative medicine are also well known and yet to be fully exploited and implemented in clinical practice [72]. All this is in support of the high hopes that novel tissue biofabrication methods and strategies can overcome many challenges in the restoration of damaged tissues lacking regenerative capacity like the joints. 

There has been a large amount of work on the 3D bioprinting of cartilage in the past two decades. The different approaches can be divided based on the choice of hydrogels (from alginate, collagen, GelMA, HAMA, PEG, etc.), biofabrication methods (extrusion-, inkjet-, or light-based), and perhaps most importantly starting cells (chondrocytes or different types of stem cells) and growth conditions. With respect to the scaffold material, key considerations, beyond the obvious ones for printability and biocompatibility, are if the ECM fibres can be pre-arranged/oriented to provide the necessary load-bearing qualities of cartilage. Another issue is the functionalisation of the matrix with growth factors or PRP as discussed. New methods like melt electrowriting and electrospinning may be able to tackle some of these structural mechanical strength and cell arrangement issues [50,73]. 

Chen et al. gave excellent examples for the importance of two frequently neglected but critical properties of the ECM—the spatial organisation of fibrils (collagen in this case) and functionalisation through ECM-bound/retained signalling moieties (stromal cell-derived factor 1 or exosomes). First, the authors found that the orientation of collagen fibres (radial/in parallel to each other) provides the scaffold with better mechanical properties (higher compressive force/displacement values) and would enhance BMSC migration in comparison to randomly oriented collagen scaffolds, especially if functionalised with SCF-1 [74]. Then, Chen et al. improved their strategy by developing a DLP method of 3D bioprinting with a mixture of cartilage-derived decellularised extracellular matrix (dECM) proteins and GelMA. This biomaterial ink was further functionalised by the successful retention of BMSC-derived exosomes. Importantly, the authors observed excellent osteochondral repair only 12 weeks after the implantation of these scaffolds in rabbits with 4 × 4 mm cylindrical knee joint lesions as judged by macroscopical ICRS (International Cartilage Repair Society) scores suggesting over 90% repair (~11 out of 12 points) and by MRI imaging. ICRS visual histological scoring also demonstrated the good quality of the resulting cartilage that filled in the lesions (16 points out of 18 maximum) [58]. It is important to note that the repair was due to endogenous chondrocytes that inhabited the 3D-bioprinted implants. Such a strategy utilising radially oriented collagen fibres with autologous ex vivo expanded chondrocytes or stem cells differentiated into chondrocytes has not been implemented yet. 

Regarding the choice of cells, it appears that so far allogenic or autologous chondrocytes may be the best choice and may be at the most advanced stage regarding the clinical application based on some excellent examples of nose-to-knee transplants [3]. Nevertheless, it is certain that research with MSCs has not reached its peak yet, and there is a lot to be expected from this area.

Lastly, the most neglected aspects of chondrogenesis in terms of the 3D biofabrication of cartilage is the way constructs are, first, cultured in vitro and second, assessed for functionality (Figure 3).

Even if there can be a debate over whether the function determines the structure or vice versa (or both are indispensable), it is well established that biomechanical stimuli greatly affect cells including chondrocytes [75]. The mechanobiology of cartilage has been extensively studied (less so during the early stages of its formation though) yet not widely implemented in 3D biofabrication protocols. Out of more than 20 biofabricated models that were discussed here, only two studies used bioreactors to improve their lab-engineered tissues. To illustrate this issue, one study applied a perfusion system to grow their 3D constructs and another group took advantage of a dynamic load device, and both found considerable improvements in cell viability and ECM secretion [46,51]. With regard to the biomechanical properties of the discussed cartilage models, none of them were particularly close to native cartilage, unless PCL or PEG was included in the scaffolds. 

The regulatory processes of bioprinting (including articular cartilage) differ around the world. The differences extend from philosophical and ethical points to issues of practical risk, biosafety, and security. Three-dimensional bioprinting techniques and materials are mostly developed in university research labs, but recently bioprinting companies have started commercialising the technology for medical purposes. The automated fabrication process could potentially produce particular defects, the risk of which needs to be specifically alleviated. There are also a number of legal issues around the liability of personalised implants regarding the processes of tissue engineering and the usage of stem cells and biomaterials.

Lastly, several largely undiscussed issues are the standardisation of manufacturing processes, the consensus on best practice guidelines, GLP and GMP compliance [76], and the overall regulation of devices, materials, and the final implants containing live chondrocytes [77]. These still unaddressed administrative and legal challenges will inevitably take a number of years to be tackled and may slow down the implementation of 3D-bioprinted articular cartilage implants in the clinic. 

## 5. Conclusions

In conclusion, in spite of a considerable amount of work and progress made in the last two decades, the 3D bioprinting of cartilage stands far from an immediate implementation in the clinic for two main reasons—the lack of standardisation of manufacturing procedures and, most importantly, the unsatisfactory biomechanical properties of the engineered tissue constructs. Nevertheless, new technologies, biomaterials, and cell culturing bioreactors will most likely speed up the development of reproducible, widely accessible, and functional 3D-bioprinted cartilage tissues that will ultimately be approved for clinical use and will provide at least partial relief to the needs of millions of patients. 

## Figures and Tables

**Figure 1 biomedicines-12-00665-f001:**
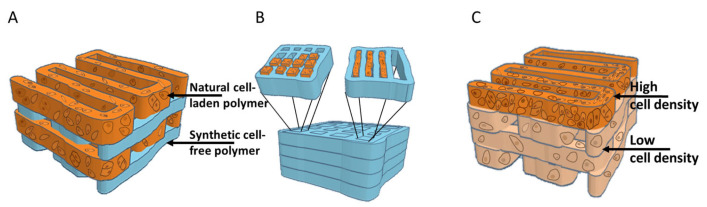
Bioprinting strategies for reconstructing the zonality and biomechanical properties of cartilage in vitro. (**A**) Alternating layers of synthetic (e.g., PCL) and natural-based bioinks to enhance the biomechanical strength of 3D-bioprinted cartilage. (**B**) The bioprinting or electrowriting of micro- or even nano-chambers or channels (with synthetic biomaterials) that can be filled in (through inkjet bioprinting for instance) with cell-laden hydrogels to recreate the zonality and biomechanics of cartilage. (**C**) The bioprinting of two or three layers with different concentrations of cells aiming to mimic the superficial and deeper zones of cartilage. The figure was constructed in tinkercad.com (accessed on the 20 February 2024) and modified in Microsoft PowerPoint (2019, version 2401).

**Figure 2 biomedicines-12-00665-f002:**
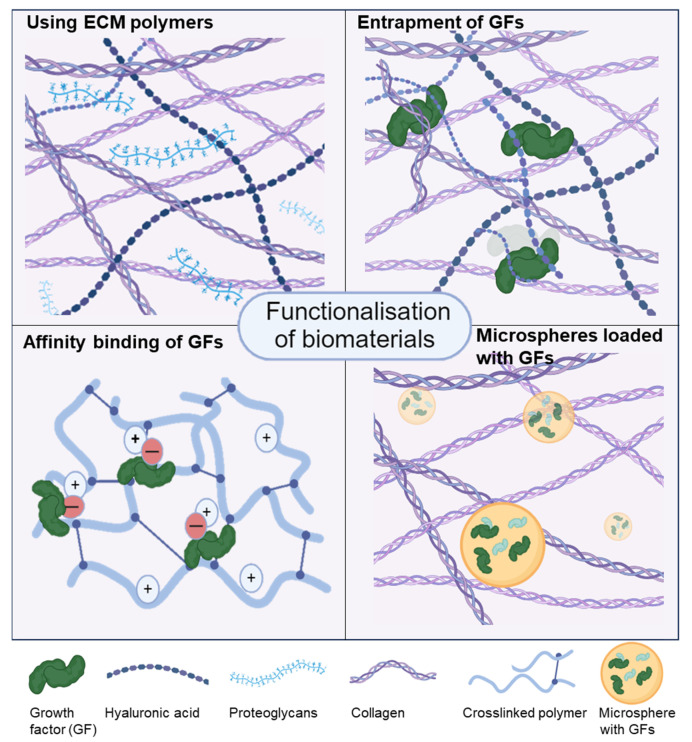
Strategies for biofunctionalisation of biomaterial inks. The figure was constructed in tinkercad.com (accessed on the 20 February 2024).

**Figure 3 biomedicines-12-00665-f003:**
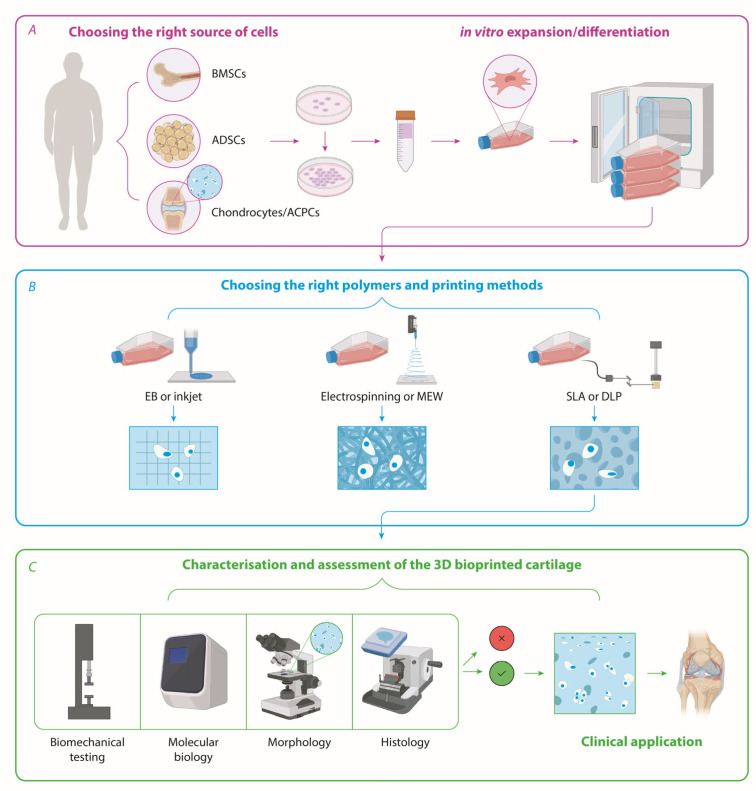
A possible workflow for the biofabrication of autologous articular cartilage implants. (**A**) The best source and type of cells are expanded (and differentiated if MSCs) in vitro; (**B**) the most appropriate biomaterial inks and 3D bioprinting methods are chosen to mimic the zonality, ECM, and biomechanical properties of native cartilage; (**C**) implants are characterised by the assessment of their biomechanical, molecular, and histomorphological properties. Legend: BMSCs—bone marrow stem cells; ADSCs—adipose-derived stem cells; ACPCs—articular cartilage progenitor cells; EB-extrusion-based; MEW—melt electrowriting; SLA—stereolithography; DLP—digital light processing. The figure was made in BioRender.com (accessed on 10 February 2024) and modified by alwaysbringatowel.com (accessed on 10 February 2024).

**Table 1 biomedicines-12-00665-t001:** Summary of current 3D-bioprinted models of cartilage.

Type of 3DBP	Type of Cells	Type of Biomaterial Ink	Key Feature	Chondrocytes and Cartilage Assessment Histological Molecular Mechanical			Ref.
EB	Equine chondrocytes, ACPCs, BMSCs	10% GelMA	Characterisation of combinations of three different cell types	Safranin O, IHC	Gene expression of chondrocyte and hypertrophy markers	Compression modulus of up to ~200 kPa at day 56 of culture	[11]
Inkjet	Human chondrocytes	PEGDA	Droplets of bioinks crosslinked by photopolymerisation	Yes (Safranin O)	no	no	[26]
EB	Human nasal septum chondrocytes	PCL/alginate	Functionalisation with TGFβ	Alcian blue, IHC	no	no	[28]
EB	Bovine chondrocytes	poly(N-isopropylacrylamide) hyaluronan (HA-pNIPAAM) and HAMA	Combination of thermosensitive and UV-crosslinkable polymers	no	no	no	[30]
EB	Primary human ADSCs (MSCs)	collagen type I (4%) and ECM granules (2.5%)	Rat dECM, human ADSCs cultured for 24 h before subcutaneous implantation in rats	Yes (Alcian blue, collagen IHC)	no	no	[36]
Inkjet and EB	Porcine chondrocytes and BMSCs	Droplets printed in PCL microchannels and GelMA	Guided spheroid formation; dynamic perfusion culture; parallel orientation of collagen fibres	Yes (Alcian blue, Safranin O, and collagen IHC)	Polarised light microscopy for collagen fibre orientation	in vitro tests (together with PCL scaffold)	[46]
EB	Equine chondrocytes, ACPCs, MSCs	10% GelMA/gellan and GelMA/gellan/HAMA (0.5%)	Two zones with ACPCs and MSCs were printed; bioprinting interfered with chondrogenesis	Safranin O staining; IHC for ECM components	Gene expression of chondrocyte markers	no	[48]
EB	Equine chondrocytes	0.5% HAMA—pHPMA-lac-PEG +/− PCL	Optimisation of concentrations according to matrix formation	Safranin O, GAG, and collagen IHC	no	in vitro Young’s modulus of 31 kPa or 4.6 MPa with PCL	[52]
EB	Rabbit BMSCs	8% alginate, 5% gelatin with or without 4% HAP	Scoring of cartilage repair and mechanical properties	Wakitani scores	no	Maximum load and compressive strength (40% restoration)	[53]
EB	hUC-MSCs	10% gelatin and 5% HAP	Knee joint lesion repair in pigs	Standard staining	Gene expression of chondrocyte markers	no	[54]
EB	Rabbit BMSCs	gelatin, fibrinogen, hyaluronic acid, and glycerol	Functionalisation with GDF5; printing together with PCL to make a scaffold	ICRS and Mankin scores	Gene expression of chondrocyte markers	in vitro tensile modulus and ultimate tensile strength (together with PCL)	[55]
EB	Rabit BMSCs	4.5% gelatin, 3% fibrinogen, 0.3% HA, and 10% glycerol; TGFβ3 and BMP4 encapsulated in PLGA; PCL scaffold	PCL/hydrogel scaffold, functionalised with TGFβ3 and BMP4 microspheres, implanted in rabbit knee joints; achieving zonal distribution of ECM	ICRS and Mankin scores; Alcian blue, Safranin O staining; immunofluorescence for ECM and chondrocyte markers	Gene expression of chondrocyte markers	Compressive modulus of ~300 kPa 12 weeks post implantation (together with PCL)	[56]
EB	ATDC5 murine chondrogenic cell line	GelMA/PRP	Photoactivation of platelets; continuous release of growth factors	Safranin O, toluidine blue, IF for collagens	Gene expression of chondrocyte markers	Storage modulus (~40 kPa)	[57]
Light-based	Porcine chondrocytes	5% GelMA or 1% HAMA	Optimisation of cell density 5–25 million/mL	Safranin O, toluidine blue, IF for collagens	Gene expression of chondrocyte markers	no	[47]
Light-based	Acellular; chondrocytes in vitro; local chondrocytes are attracted in vivo	10% GelMa, 2% decellularised ECM, and BMSCs exosomes	Radial orientation of channels; exosome functionalisation	ICRS macroscopic and histological scores	Proteomics	no	[58]
Light-based	Human BMSCs	10% GelMA, 5% PEGDA, TGF-β1 nanospheres	Mechanical strength with PEGDA and functionalisation with TGF-β1	Safranin O and Alcian blue	Gene expression of chondrocyte markers	In vitro compressive modulus (~MPa at 5% PEGDA)	[59]
MEW	Equine chondrocytes	PCL scaffold; 10% GelMA	Tangential orientation of PCL fibres; growth under dynamic compression bioreactor	Safranin O, IHC	no	12-fold increase in peak modulus under congruent loading when GelMA was reinforced (~500 kPa)	[51]
MEW and inkjet	Porcine BMSCs	PCL scaffold; droplets	Self-organisation of collagen fibres secreted by spheroids into zones	Alcian blue, Safranin O, IHC	no	Compressive modulus of ~400 kPa; equilibrium modulus of ~200 kPa, and dynamic modulus of ~2.6 MPa	[50]

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
