# Peer review of "High Hopes for the Biofabrication of Articular Cartilage—What Lies beyond the Horizon of Tissue Engineering and 3D Bioprinting?"

_biomedicines, 2024, doi:10.3390/biomedicines12030665_

Round 1

Reviewer 1 Report

Comments and Suggestions for Authors

In this review, the authors report and describe the chondrocytes and biofabrication of articular cartilage. The review is comprehensive and well written, describing the basic facts about chondrocytes, their sources and 3D printing technology, connected to chondrocytes and cartilage.

The abstract is well written and informative. The Introduction offers basic insights into chondrocytes. Noting to add. Furthermore, sources for cartilage and bioprinting possibilities are described. The limitations are briefly discussed as well.

Line 64: The authors, according to the tile, may describe the sources for chondrocyte isolation.

Line 144: I do not understand the meaning of this sentence. Please, clarify or rewrite.

I do not have to add anything particular, the text s well written and this review is suitable for publication in this journal in its present form. There is nothing to add or change, except the above mentioned questions.  

Comments on the Quality of English Language

Minor editing of English language recommended. 

Author Response

Dear Reviewer, 
Thank you for taking the time to critically assess our manuscript. We appreciate all comments and have taken care of the issues with the text you raised. 

Reviewer 2 Report

Comments and Suggestions for Authors

The manuscript presented by Sbirkov et al. is a review in the field of cartilage tissue engineering. This review provides a concise overview of recent advances in the field. Overall, the manuscript is well written and structured. However, in my opinion, the manuscript is poor in illustrative material. The following revisions should be made:

1.    A few figures should be added. In the review, it is acceptable to borrow figures from original studies (with the permission from the publisher, if they are not in the open access) or adapt them. For example, a figure illustrating a technique for creating zonality would be very useful. In addition, a figure summarizing the structures of the polymers used, examples of materials with excellent mechanical properties, possible designs or illustrations for possible biofunctionalization can be added.

2.    The manuscript contains only 1 figure and there is no reference to this figure in the text. Please find a place for figure 1 in the text.

3.    Figure should be placed after the paragraph where it is first mentioned.

Author Response

Dear Reviewer, 
Thank you very much for the constructive comments! We have prepared 2 additional figures as suggested and believe that the manuscript has improved substantially now.  

Round 2

Reviewer 2 Report

Comments and Suggestions for Authors

The manuscript has been revised in accordance with the requested issues. The manuscript can be accepted in its current form.